# THE FREE TRANSFORMER

## ABSTRACT

We propose an extension of the decoder Transformer that conditions its generative process on random latent variables. Those variables are learned without supervision thanks to a variational procedure. Experimental evaluations show that allowing such a conditioning translates into substantial improvements on downstream tasks.

## 1 INTRODUCTION

Since their invention, the Transformer (Vaswani et al., 2017), and more specifically the decoder-only Transformers used originally for the GPT series of models (Radford et al., 2018), have become the core components of AI systems.

It is remarkable that, after almost a decade, and in spite of improvements on many aspects of this class of methods, the autoregressive modelling of Transformers remains essentially unchallenged. We propose in this paper to revisit this key design aspect by allowing richer and more natural density models to emerge:

- We extend the auto-regressive model of the decoder Transformer by allowing the conditioning on latent variables, thanks to a formulation as a conditional Variational Autoencoder (§ 3.1).
- We propose an implementation that requires a very modest computational and memory usage overhead (§ 3.2).

The benefits of the proposed method are shown by training 1.5B and 8B models from scratch and assessing performance on multiple downstream benchmarks (§ 4).

## 2 MOTIVATION

Decoder Transformers are auto-regressive discrete density approximators. They model a sequence of tokens $S_1, \ldots, S_T$ by estimating the conditional distribution of each given those preceding it. Sampling is done by generating one token after another, each time computing the distribution of the next symbol given those generated so far.

The only density modelling and sampling that such models implement is that of the generated tokens. In particular, a decoder Transformer does not make additional latent decisions about the stream of symbols to generate. Its only decisions are the choices of the tokens themselves.

Consider, for instance, that we train such a model to generate movie reviews and that we want to have two clearly separated categories of negative and positive reviews. Given a large enough model and the necessary amount of training data, there is no doubt that a decoder Transformer trained on a dataset of that form would work perfectly and would generate these two types of reviews. However, to do so, it would generate tokens one after another and decide, based on the words generated so far, whether the review it is currently generating is a positive or a negative one, and continue the process accordingly. In particular, *the model would not make the explicit decision to generate a negative or a positive review*. It would produce tokens, and this notion of a negative or positive review would be implicit in their posterior probabilities.

Due to the chain rule, any density can be modelled as autoregressive. However, in particular when the "natural" structure involves conditioning on latent variables, the autoregressive model of the signal may be a great deal more complex than the full joint model including the latent.

We can consider a simple example illustrating that point. Let $Z \sim \mathcal{B}(0.5)$ be a latent "coin flip", and $X_1, \ldots, X_T$ be equal to $Z$ with independent flips of probability $\epsilon$.

The $X_t$s are conditionally independent given $Z$, and we have

$$P(X_t = 1 \mid Z = z) = \epsilon z + (1 - \epsilon)(1 - z) \tag{1}$$

however, expressed as an auto-regressive model without $Z$, we get:

$$P(X_{t+1} = 1 \mid X_1 = x_1, \ldots, X_t = x_t) = \frac{\left(\frac{\epsilon}{1-\epsilon}\right)^{\sum_{s=1}^{t} x_s} (1 - \epsilon)^t \epsilon + \left(\frac{1-\epsilon}{\epsilon}\right)^{\sum_{s=1}^{t} x_s} \epsilon^t (1 - \epsilon)}{\left(\frac{\epsilon}{1-\epsilon}\right)^{\sum_{s=1}^{t} x_s} (1 - \epsilon)^t + \left(\frac{1-\epsilon}{\epsilon}\right)^{\sum_{s=1}^{t} x_s} \epsilon^t}. \tag{2}$$

We could easily come with worse examples when expressed autoregressively, for instance when the latent variables are positions in the sequence, e.g. indexes where certain patterns occur as in the example of § 4.1. What we observe in such cases is that it requires running estimates of probabilities ("probability that the target appears here") for which estimation errors are unavoidable and problematic.

The consequence is that a purely auto-regressive density model suffers potentially from several drawbacks:

- It requires an unnecessarily complicated computation, and greater capacity, to implicitly make post-hoc decisions or infer latent quantities from the generated tokens.

- It may be sent off track during the process if, by mistake, a few tokens generated are erroneous, ambiguous or contradictory with those generated previously.

- Key concepts do not appear spontaneously due to the "natural" factorization of the distribution, but are built post-hoc by necessity to fit the training samples better. This may be a fundamental weakness when operating out of distribution.

The main objective of the present work is to address these issues by providing the model with the freedom of conditioning its auto-regressive process on latent random quantities that are not imposed by the training examples.

For instance, for the review generator example above, the model could use a random Boolean value to decide once for all whether the tokens it produces are from the distribution of negative or positive reviews, removing the need for a complicated posterior estimate from the tokens already generated.

## 3 METHOD

Any latent random value $Y_r$, whatever its statistical dependency with the tokens $S_1, \ldots, S_t$ and other latent $Y_1, \ldots, Y_{r-1}$ sampled so far, can be expressed under reasonable assumptions as $f_r(S_1, \ldots, S_t, Y_1, \ldots, Y_{r-1}, Z_r)$ where $Z_r$ is a value coming from a random generator.

Hence, if we provide the model with enough random values $Z_1, Z_2, \ldots$ sampled independently during generation, a proper training procedure could in principle build families of latent variables with arbitrary dependency structure, as long as the model's capacity allows it to encode $f_r$.

In the same way that the choice of a token during sampling can be expressed as a function of a random value and the logits, any activation which is a function of a random value and other activations can be interpreted as a decision made by the model during the generative process. Such decisions make the latent activation non-deterministic functions of the tokens, and observing the latter only gives a partial information about the former.

### 3.1 CONDITIONAL VARIATIONAL AUTOENCODER

Generating a full sequence from scratch with a model that depends on a random variable $Z$ is trivial: sample $Z \sim P(Z)$ and then run the standard auto-regressive process, with the computation of the logits modulated by $Z$.

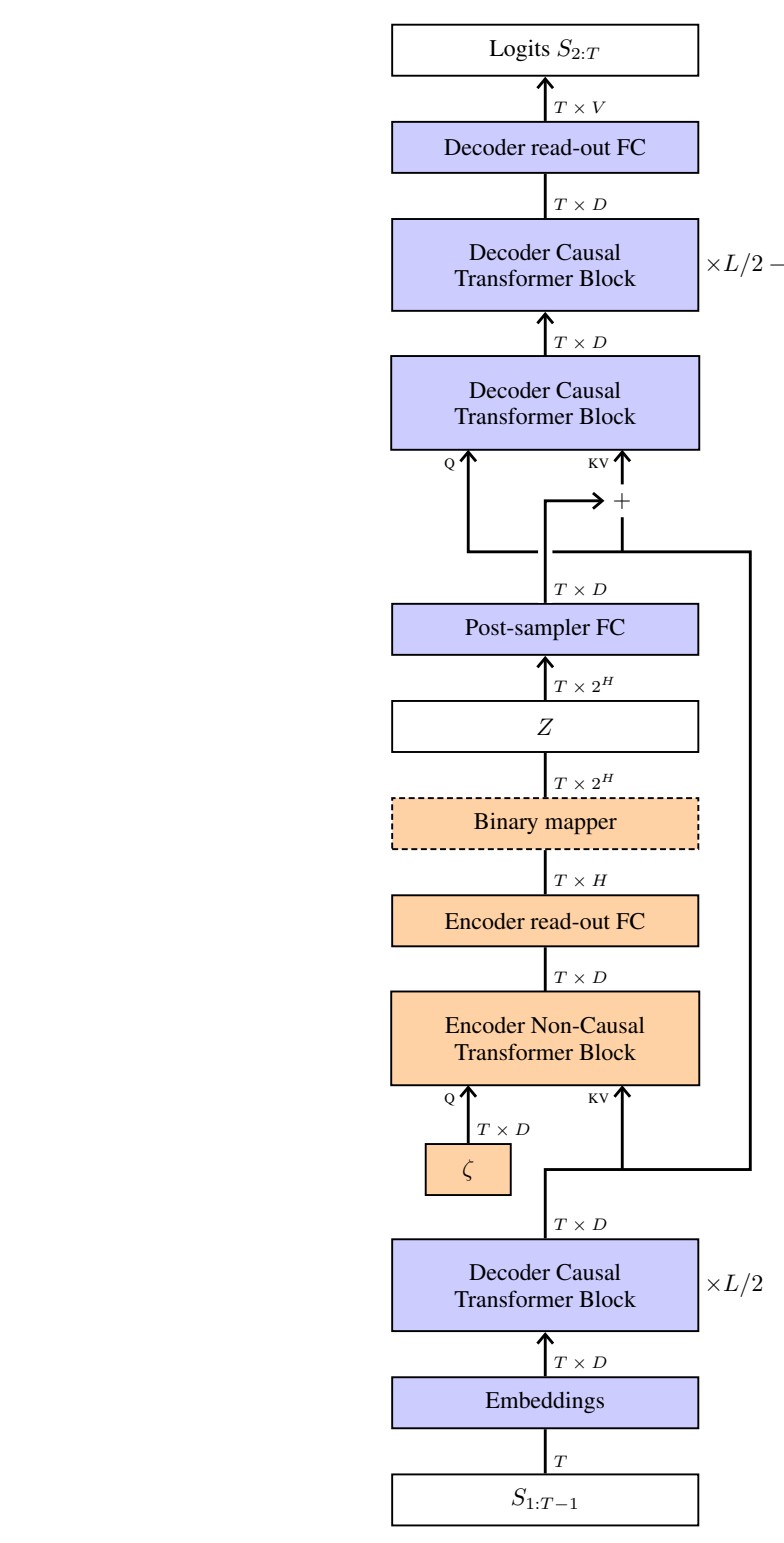

Figure 1: The Free Transformer. We omit the normalization layers and residual connections from the model and the batch size from the tensor shapes for clarity. The operators in orange are specific to the encoder and are evaluated only for training or KV cache pre-filling, those with a dashed contour have no trainable parameters. The Binary Mapper is described in § 3.4. During generation, the encoder is not evaluated and $Z$ is sampled uniformly among the one-hot vectors of dimension $2^H$.

Training the model, however, is far more involved. Given a training sample $S$, the objective is to maximize

$$P(S) = \int_z P(S \mid Z = z)P(Z = z)dz, \tag{3}$$

which can be estimated only if we can get $Z$s consistent with $S$, that is *$Z$s that we would sample if the overall process was generating $S$*. This amounts to a complex inference problem if we want $Z$ to capture meaningful structural properties of the sequence.

Providing those $Z$s is the role of the encoder of a Variational Autoencoder (Kingma & Welling, 2013), whose main purpose is to sample from a "good" distribution $Q(Z \mid S)$ so that a sampled $Z$ modulates the decoder in a way that leads it to generate $S$.

We follow this approach and optimize jointly the parameters of the decoder and the parameters of a second model, which is an encoder in the VAE sense.

Even though the noise $Z$ has no relation to $S$ initially, if the training succeeds, the model will use it to structure the generative process. In the example of a movie review generator of the previous section, for instance, given a review from the training set, the encoder would implicitly classify it as positive or negative, and generate a consistent $Z$. Increasing $P(S \mid Z)$ with that $Z$ could be interpreted as improving the "negative review generator" or the "positive review generator" that are implicitly encoded in the decoder's weights.

A key element of this approach is to limit the amount of information flowing from the encoder to the decoder through $Z$, so that the encoder does not provide quantities that should be computed by the decoder. At the limit the encoder could copy entirely $S$ into $Z$ so that a trivial decoder, useless without the encoder, hence in inference, would score perfectly in training.

The formal derivation of the VAE shows that the proper measure of information is the Kullback-Leibler divergence between $Q(Z \mid S)$ and $P(Z)$, and that the loss to minimize should sum it with the reconstruction loss, which here is the usual cross-entropy.

### 3.2 MODEL STRUCTURE

In what follows, we call "Transformer Block" the usual combination of a Multi-Head Attention layer and a MLP-like tokenwise module, with normalisation layers and residual connections.

As pictured on Figure 1, the Free Transformer is a standard decoder with a noise $Z$ injected in its middle layer. This allows to share half of the Transformer blocks of the decoder with the encoder, cutting down drastically the computational overhead by having a single Transformer block that has to be computed specifically for the encoder. Hence, as we will see, this model possesses all the components of a decoder Transformer and has an additional non-causal block and two linear layers for the encoder. While we did not investigate what is the best depth to inject $Z$, doing it too early would reduce the encoder's capacity, and doing it too late would reduce the decoder's capacity to process the latent variables.

For clarity, we omit in what follows the batch size in the tensor shapes.

As a standard decoder Transformer, the Free Transformer processes a sequence of tokens by first encoding them with the embedding table into a tensor $X_0$ of shape $T \times D$.

Then it evaluates sequentially the first $L/2$ Transformer blocks to get $X_{L/2}$ of same shape, and at this point, it samples a sequence of one-hot vectors $Z = (Z_1, \ldots, Z_T) \in \{0, 1\}^{T \times C}$. During generation, this is done by sampling, for each $Z_t$, an index $c$ uniformly in $\{0, \ldots, C - 1\}$, and then encoding it as a one-hot vector of dimension $C$. During training or KV cache pre-filling, $Z$ has to be consistent with the tokens of $S$ already fixed, and the sampling is done with the encoder instead, as described in § 3.3.

This tensor $Z$ is processed by a linear layer to obtain a tensor $R$ of shape $T \times D$. Then, the $L/2+1$th Transformer block gets as input for queries the tensor $X_{L/2}$ and as input for keys and values the tensor $X_{L/2} + R$. The rest of the Transformer blocks are evaluated in sequence to get $X_L$ which is processed by the read-out linear layer to obtain the logit tensor $L$ of shape $T \times V$, where $V$ is the vocabulary suze.

### 3.3 ENCODER AND LOSS

As stated in the previous section, during training or KV cache pre-filling, the tensor $Z$ is sampled with the encoder.

The Free Transformer possesses one Transformer block specific to the encoder, which is non-causal, making the encoder as a whole non-causal. This is necessary since the conditioning by the decoder may have long-range effects, requiring the full sequence to be taken into account to get a proper conditional distribution of the latent.

This encoder-specific block gets as input for the queries a trained token embedding $\zeta$ replicated to match the sequence length, and for the keys and values the output of the first half of the decoder's blocks. The motivation for using a learned constant input for the queries instead of the standard representation of the input sequence is to prevent the encoder from building a token-wise mapping and make it instead capture global properties of the sequence that may be more transferable across tasks and data-sets.

A linear readout computes from the encoder block's output a vector of dimension $H = 16$ for every token. These components are interpreted as logits of individual bit, used to sample a value in $\{0, \ldots, 2^H - 1\}$ which is encoded into a one-hot vector of dimension $2^H = 65,536$, with gradient pass-through, as described in § 3.4.

Hence, the random embedding $Z$ is a sequence of $T$ one-hot vectors $Z_t$ of dimension $2^H$. The prior distribution used for generation is uniform $P(Z_t = z) = 1/2^H$, and $Q(Z \mid S = s)$ is the distribution corresponding to the sampling with the encoder described above. The KL divergence is then equal to

$$\mathbb{D}_{\text{KL}}\Big(Q(Z_t \mid S_1, \ldots, S_T) \,\Big\|\, P(Z_t)\Big) = H \log 2 + \sum_{z=1}^{2^H} Q(Z = z \mid S) \log Q(Z = z \mid S). \quad (4)$$

We control it by adding it to the loss, and prevent its collapse by using a token-wise free bits method (Kingma et al., 2016). This means that we sum the KL divergence of individual $Z_t$ that are above a threshold $\kappa$ and ignore the others.

This leads us to use for training loss the sum of the standard cross-entropy and the following quantity

$$\frac{1}{T} \sum_{t=1}^{T} \max\Big(0, \mathbb{D}_{\text{KL}}\Big(Q(Z_t \mid S_1, \ldots, S_T) \,\Big\|\, P(Z_t)\Big) - \kappa\Big), \quad (5)$$

where the threshold $\kappa$ is an hyperparameter.

### 3.4 BINARY MAPPER

The last linear layer of the encoder computes for every index $t$ of the sequence being processed a vector $L_t = (L_{t,1}, \ldots, L_{t,H}) \in \mathbb{R}^H$, whose components are interpreted as the logits of individual bits of a binary encoding.

The Binary Mapper samples those bits $B_{t,1}, \ldots, B_{t,H}$ independently with

$$P(B_{t,h} = 1) = \frac{1}{1 + e^{-L_{t,h}}}, \quad (6)$$

and outputs a one-hot vector $Y_t$ of dimension $2^H$ corresponding to the resulting value:

$$Y_{t,d} = \begin{cases} 1 & \text{if } d = 1 + \sum_{h=1}^{H} 2^{h-1} B_{h,t} \\ 0 & \text{otherwise.} \end{cases} \quad (7)$$

During training, the computation also propagates the gradient of the probabilities of the $2^H$ values. If $U(d) = (U_1(d), \ldots, U_H(d)) \in \{0, 1\}^H$ is the binary encoding of $d$, and we define $G_t$ as

$$G_{t,d} = P(B_t = U(d-1))$$

$$= \exp\left(\sum_h \log P(B_{t,h} = U_h(d-1))\right)$$

$$= \exp\left(\sum_h (1 - U_h(d-1)) \log\left(1 - \frac{1}{1 + e^{-L_{t,h}}}\right) + U_h(d-1) \log\left(\frac{1}{1 + e^{-L_{t,h}}}\right)\right),$$

then the Binary Mapper outputs

$$Y_{t,d} + G_{t,d} - \text{detach}(G_{t,d}), \tag{8}$$

where $\forall x, \text{detach}(x) = x$ and $J_{\text{detach}}(x) = 0$.

The motivation for using a binary encoding instead of having the encoder output $2^H$ logits directly is to facilitate the gradient pass-through thanks to the monotonicity of the sigmoid.

## 4 EXPERIMENTS

We first test the qualitative behavior of the Free Transformer on a synthetic task in § 4.1, then compare it on multiple benchmarks to baselines with 1.5B and 8B parameters models for various KL divergence thresholds in § 4.4, and finally assess the performance gain of a 8B parameter model trained on 1T tokens in § 4.5.

### 4.1 SYNTHETIC DATASET

To confirm that the Free Transformer indeed utilizes $Z$ to condition its generative process, we designed a synthetic dataset and trained a small Free Transformer with different free-bits thresholds. Doing so allows to observe what aspects of the modeling are packed by the encoder in $Z$.

Each sequence in our synthetic training set is generated as follows:

- start with a string of 64 underscores "_",
- pick an upper case letter and a position in the sequence at random, and replace the underscores there with a "target" made of the selected letter repeated 8 times,
- replace any character with an exclamation mark with probability $1/16$
- concatenate a prompt made of the target's letter followed by a ">".

A few sequences generated with that process are shown in Figure 2, Appendix B.

We trained a Free Transformer on this data for four different values of the free bits threshold $\kappa$, and generated with the same random prompt three groups of sequences with each model, as pictured in Figure 3, Appendix B. For each model, in the blue group, the noise $Z$ is sampled independently for each sequence, whereas we sampled one $Z$ only for each of the green groups, used to generate all its sequences.

For very low values of the KL divergence, the model behaves like a vanilla model (Figure 3, top left), and when the value increases, the model encodes initially the position of the target alone in the latent state (Figure 3, top right), then encodes both the target position and the noise (Figure 3, bottom left), and finally encodes the full sequence, resulting in incorrect generation (Figure 3, bottom right).

### 4.2 BASELINE ARCHITECTURES

For assessing performance on standard benchmarks we used decoder-only Transformers implemented in a sota proprietary Transformer codebase. Those are well optimized models using the SwiGLU non-linearity (Shazeer, 2020), pre-normalization with RMSNorm (Zhang et al., 2019), Rotary Positional Embedding (RoPE, Su et al. 2021), and Group Query Attention (GQA, Ainslie et al. 2023). The vocabulary size is $2^{17} \approx 130k$.

We used two sizes of models:

- A 1.5B model, with 28 layers, weight tying between the embeddings and the logit readout, model dimension 1536, 12 query heads, and 2 key-value heads. It is trained with 47B tokens, which requires 32 H100s for $\approx 12$ hours.

- A 8B model with the structure of a Llama-3, which is 32 layers, model dimension 4096, 32 query heads, and 8 key-value heads. It is trained with 200B tokens which requires 256 H100s for $\approx 24$ hours, or with 1T tokens, which takes 5 days.

We compare those baselines to the equivalent Free Transformers, which require one additional layer for the encoder during training and KV cache pre-filling, resulting in a compute and memory overhead of $1/28 \approx 3.6\%$ for the 1.5B and $1/32 \approx 3.1\%$ for the 8B.

### 4.3 Setup and hyperparameters

We kept our findings as clear as possible by avoiding other sources of performance improvement:

- We stuck to the baseline architecture, optimizer, and learning rate schedule that were used to train the baselines, and did not optimize any hyperparameter for our setup.

- We avoided any recipes for the VAE components, such as removing sampling in inference. We followed the formal expressions rigorously.

- We fixed $H$ to 16 so that the dimention of $Z_t$ was comparable to the vocabulary size of $2^{17}$.

We stress that the optimization hyperparameters were highly tuned for the baselines, and it is probable that a combination of an encoder and a decoder has specific requirements that would greatly benefit from an adapted training procedure.

### 4.4 Exploratory Results

We ran a series of experiments to assess the general behavior of the Free Transformer, and to calibrate the $\kappa$ threshold.

For any value of $\kappa$, the cross-entropy goes down regularly during training, with no more instability and spikes than what happens with the baselines. The KL divergence rapidly goes under $\kappa$ and stays there. When we compare the cross-entropies for various $\kappa$, they go down when $\kappa$ increases as expected, but the values remain extremely close, with a difference of the order of $0.01$ for a cross-entropy of $\approx 2$ for the 1.5B and $\approx 1.8$ for the 8B.

For both sizes of models, setting $\kappa = 4 \log 2$, corresponding to 4 bits of information per token, resulted in a collapse of the cross-entropy, indicating that the encoder found a way to channel fully the tokens to predict, and resulting in a collapse of performance on the downstream tasks. It is noteworthy that the baseline 8B model reaches during training a cross-entropy of $1.8 = 2.59 \log(2)$, hence may explain why allowing 2 bits does not collapse, while allowing 4 bits does.

The performance on downstream tasks are given in Appendix E, Table 2 for the 1.5B models, and Table 3 for the 8B models, both for four different values of $\kappa$ corresponding to $1/2$ to 2 bits of information per token. Graphs of performance during training are given in Appendix F in Figures 4 and 5.

We observe a substantial increase of performance on HumanEval+, MBPP, and GSM8K which are arguably the benchmarks requiring some form of reasoning, and there also is a clear improvement for the 8B model with $1/2$ bit of KL divergence on MMLU and CSQA, which are multi-choice questions.

### 4.5 Results with 1T tokens training

To measure improvement in a more realistic setting, closer to models actually used in real applications, we trained 8B models on 1T tokens, which improves drastically the performance of both the baseline and the Free Transformer.

| 8B models (1T tokens) | | | | | | |
|---|---|---|---|---|---|---|
| | **Final value** | | | **Average (last third)** | | |
| | **Baseline** | **Free Transformer** 1/2 bit | | **Baseline** | **Free Transformer** 1/2 bit | |
| Generative code/math | | | | | | |
| human_eval_plu (pass@1) | 0.268 | 0.299 | +11.36% | 0.245 | 0.256 | +4.22% |
| mbpp (pass@1) | 0.428 | 0.440 | +2.80% | 0.396 | 0.421 | +6.08% |
| gsm8k (em) | 0.321 | 0.331 | +2.83% | 0.280 | 0.296 | +5.84% |
| Multi-choice general knowledge / common sense | | | | | | |
| mmlu (macro_avg/acc_char) | 0.592 | 0.623 | +5.20% | 0.567 | 0.596 | +5.16% |
| csqa (acc_char) | 0.707 | 0.748 | +5.79% | 0.689 | 0.733 | +6.28% |
| hellaswag (acc_char) | 0.799 | 0.799 | -0.01% | 0.787 | 0.788 | +0.18% |
| winogrande (acc_char) | 0.739 | 0.735 | -0.53% | 0.725 | 0.727 | +0.27% |
| obqa (acc_completion) | 0.564 | 0.562 | -0.35% | 0.556 | 0.551 | -0.86% |
| arc_challenge (acc_completion) | 0.542 | 0.535 | -1.42% | 0.524 | 0.522 | -0.40% |
| arc_easy (acc_completion) | 0.721 | 0.711 | -1.41% | 0.706 | 0.711 | +0.68% |
| piqa (acc_char) | 0.805 | 0.812 | +0.88% | 0.802 | 0.807 | +0.61% |
| Multi-choice text understanding | | | | | | |
| race.high (acc_char) | 0.473 | 0.463 | -2.06% | 0.467 | 0.460 | -1.55% |
| race.middle (acc_char) | 0.632 | 0.634 | +0.33% | 0.623 | 0.624 | +0.16% |
| boolq (acc_completion) | 0.713 | 0.725 | +1.63% | 0.755 | 0.754 | -0.10% |
| Culture | | | | | | |
| nq (em) | 0.248 | 0.247 | -0.22% | 0.229 | 0.227 | -0.76% |
| tqa (em) | 0.583 | 0.577 | -1.00% | 0.549 | 0.544 | -0.90% |

Table 1: Performance of 8B models trained on 1T tokens. We also provide the average over the last third of the iterations to mitigate the irregularity of the performance increase during training and get a more accurate estimate of the relative improvement. The optimization hyperparameters were tuned for the baseline and kept unchanged, but the Free Transformers require 3.1% more compute and parameters for the encoder. See Figure 6 in Appendix F for the performance during training.

Given the results with 200B tokens, we chose the value $\kappa = \log(2)/2$ corresponding to half a bit of information per token at most.

The performance on downstream tasks are given in Table 1 and the corresponding graphs during training in Figure 6 of Appendix F. We provide in the table the performance measured at the end of the training as for the other configurations, but in addition we also give the average over the last third of the training. We can observe on the graphs that the rate of improvement tend to be constant on this interval, which justifies averaging to mitigate the performance fluctuations.

The key result is the boost of performance on HumanEval+, MBPP, GSM8K, MMLU and CSQA, confirming what we observed in the smaller settings, and a greater stability on other tasks.

## 5 PREVIOUS WORK

There have been several attempts at combining a VAE and a decoder Transformer, generally with a focus on improving topic models and providing ways to guide the generation.

The OPTIMUS model (Li et al., 2020) combines a pre-trained BERT as text embedding / encoder, with a GPT-2 playing the role of decoder, which are fine-tuned with a VAE-like loss.

The latent embedding $Z$ is computed thanks to a CLS token, that is by adding a token to the input and a read-out to extract its embedding in the output. To modulate the GPT-2 generation with it, it is either (1) concatenated as an additional token in every layer, or (2) added to the input token embeddings. Collapse of the KL divergence is prevented during training with the free bits method (Kingma et al., 2016).

This approach allows for better guided text generation with GPT-2 and better generalization on low-data languages with BERT.

Xie et al. (2021) extend OPTIMUS with a multi-objective loss, adding in particular the prediction of the story topic, using the output of another model as ground truth, to obtain a better embedding space.

The CVAE proposed by Fang et al. (2021) combines two pre-trained GPT-2, one used as the encoder without causal masking. The embedding $Z$ is an average of the encoder's output, and the authors propose three ways to modulate the decoder with linear images of it: (1) add it to each input token embedding, (2) concatenate it to the Ks and Vs in every layer, (3) add it before the softmax. Experiments demonstrate that this method allows controlling the generation without hurting the quality of the result.

AdaVAE (Tu et al., 2022) is similarly the combination of two pre-trained GPT-2, the first without causal masking playing the role of the encoder. The latent embedding $Z$ is extracted from its output with a slightly modified attention operator. It is then injected into the decoder by either concatenating an image of it to the keys and values as in OPTIMUS, or before the softmax as in CVAE.

## 6 CONCLUSION

The Free Transformer is a direct extension of a standard decoder Transformer, with the abstract structure of a conditional VAE. It is implemented with a single additional non-causal Transformer block and requires a few percent of computational and memory usage overhead.

Its structure makes it able to learn latent random variables unsupervised, and to condition its generative process on them. In some ways, this approach aims at achieving in latent space with an autoencoder what reasoning models do with chains-of-thought in token space and an RL procedure (DeepSeek-AI et al., 2025). A combination of the two is, of course, promising.

The performance boost without tuning the optimization hyperparameters across multiple benchmarks and two sizes of models, is a strong signal that the overall approach actually improves the inductive bias of the vanilla Transformer.

Many properties and design choices should be explored. The performance curves during training are often unstable, possibly due to the coupling of the optimization of the encoder and the decoder,

and using different optimization methods could be fruitful. The random embedding itself could take many forms, and the one used in our implementation is arbitrary.

Finally, the behavior in larger scales, both in parameter count and dataset size, remains to be investigated.

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

## A    ALGORITHMS

---

**Algorithm 1** Forward pass of a standard decoder Transformer

---

1: **procedure** FORWARD($tokens$)
2:     $x \leftarrow$ embeddings($tokens$)
3:     **for** $n = 1, \ldots, B$ **do**
4:         $x \leftarrow$ blocks$[n](in = x)$
5:     **end for**
6:     $logits \leftarrow$ linear_readout(RMS_norm($x$))
7:     **return** $logits$
8: **end procedure**

---

**Algorithm 2** Forward pass of a Free Transformer

---

1: **procedure** FORWARD($tokens$)
2:     $x \leftarrow$ embeddings($tokens$)
3:     **for** $n = 1, \ldots, B/2$ **do**
4:         $x \leftarrow$ blocks$[n](in = x)$
5:     **end for**
6:     **if** $train$ or $prefill$ **then**
7:         $y \leftarrow$ encoder_block($in\_q = zeta, in\_kv = x$)
8:         $o \leftarrow$ encoder_linear_readout(RMS_norm($y$))
9:         $z \leftarrow$ binary_mapper($o$)
10:    **else**
11:        $z \leftarrow$ one_hot(uniform_sampler())
12:    **end if**
13:    $r \leftarrow$ linear_post_sampler($z$)
14:    $x \leftarrow$ blocks$[B/2 + 1](in\_q = x, in\_kv = x + r)$
15:    **for** $n = B/2 + 1, \ldots, B$ **do**
16:        $x \leftarrow$ blocks$[n](in = x)$
17:    **end for**
18:    $logits \leftarrow$ linear_readout(RMS_norm($x$))
19:    **return** $logits$
20: **end procedure**

---

# B  SYNTHETIC EXPERIMENT

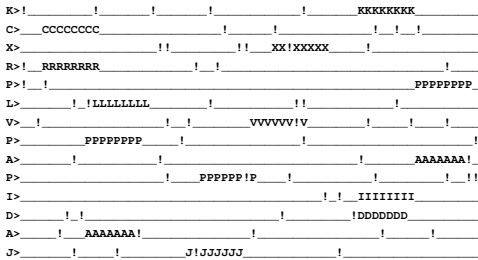

Figure 2: The synthetic sequences of § 4.1 are of fixed length, with a "target" made of a random letter repeated 8 times at a random position, an i.i.d. noise of exclamation marks, and a prompt indicating the target's letter.

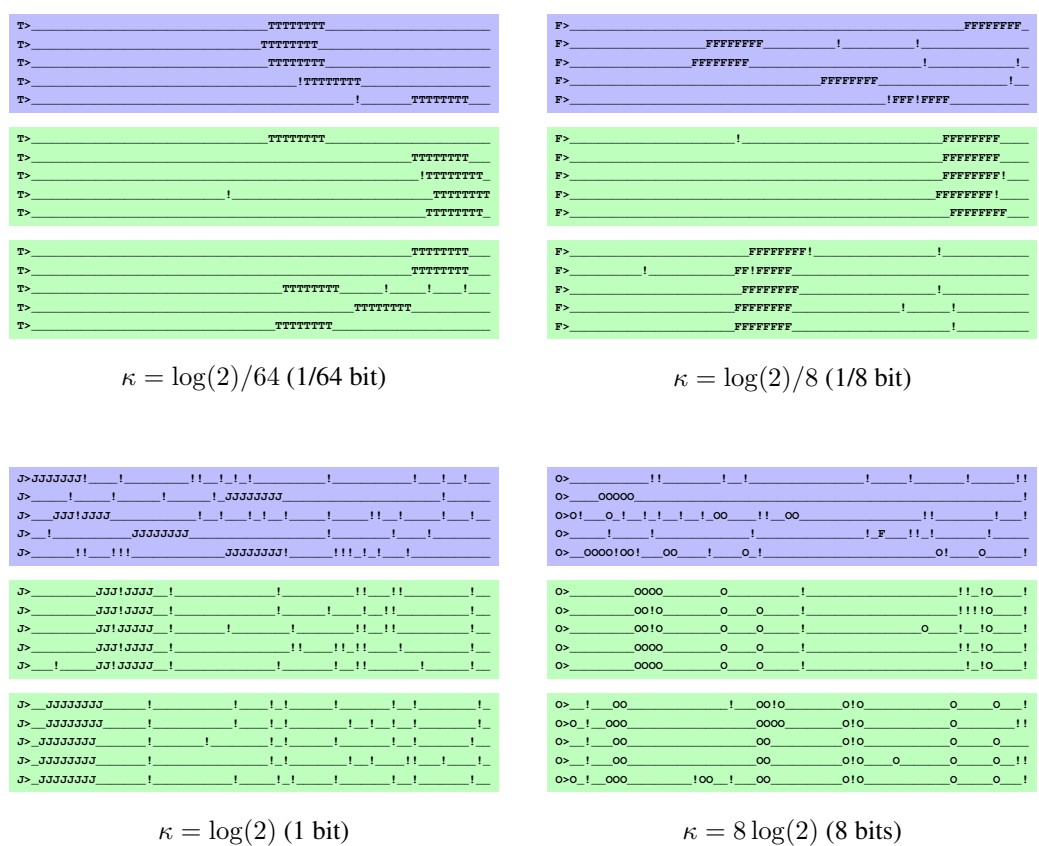

Figure 3: Results with a Free Transformer trained on the synthetic sequences of § 4.1 for different prompts and free bit thresholds. To investigate the information encoded in the latent tensor, we sample a $Z$ per sequence of a blue box, and a $Z$ per green box. For very low values of the KL divergence, the model behaves like a vanilla model (top left), and when the KL divergence increases, the model encodes initially the position of the target alone in the latent state (top right), then encodes both the target position and the noise (bottom left), and finally encodes the full sequence, resulting in incorrect generation (bottom right).

## C    EVALUATION BENCHMARKS

- HellaSwag: Multiple choices. Common sense focusing on physically situated scenarios. (Zellers et al., 2019)
- WinoGrande: Large-scale adversarial Winograd-style pronoun resolution (fill-in-the-blank) designed to reduce annotation artifacts. (Sakaguchi et al., 2019)
- ARC (AI2 Reasoning Challenge): Grade-school science multiple choice. (Clark et al., 2018)
- PIQA: Physical commonsense multiple choice about everyday goals and affordances. (Bisk et al., 2019)
- OpenBookQA (OBQA): Open-book science QA: combines a provided set of core facts with commonsense/world knowledge to answer questions. (Mihaylov et al., 2018)
- RACE: Multiple-choice reading comprehension from Chinese middle-school English exams. (Lai et al., 2017)
- MMLU: "Massive Multitask Language Understanding". Questions spanning STEM, humanities, social sciences, etc. (Hendrycks et al., 2021)
- CommonsenseQA (CSQA): Multiple-choice QA requiring commonsense relational knowledge (leveraging ConceptNet relations). (Talmor et al., 2019)
- BoolQ: Yes/no questions paired with passages to evaluate reading comprehension and entailment-like inference. (Clark et al., 2019)
- GSM8K: Grade-school math word problems requiring multi-step arithmetic reasoning. (Cobbe et al., 2021)
- HumanEval+: An augmented version of OpenAI's HumanEval (Chen et al., 2021) with many more unit tests per problem to reduce test fragility and overfitting in code generation evaluation. (Liu et al., 2023)
- MBPP: "Mostly Basic Programming Problems." Short Python programming tasks solvable by entry-level programmers; includes text spec and example tests. (Austin et al., 2021)
- NQ: "Natural Questions." Real user queries paired with Wikipedia pages. (Kwiatkowski et al., 2019)

## D    PERFORMANCE MEASURES

- For generated answers:
  - pass@1 is the proportion of generated pieces of code that produce the expected behavior when executed.
  - em ("exact match") is the proportion of generated endings of a sequence that perfectly match a reference solution.
- For multi-choice based on log probabilities:
  - acc_completion is the proportion of correct responses when the choice is based on the sum of the log probabilities normalized with the number of tokens of each possible choices.
  - acc_char is the same as acc_completion but normalizes with the number of characters.
  - macro_avg/acc_char is the average of acc_char over multiple sub-categories of questions.

# E EXPLORATORY PERFORMANCE

| **1.5B models (47B tokens)** | | | | | | | | |
|---|---|---|---|---|---|---|---|---|
| | **Baseline** | **Free Transformer** | | | | | | |
| | | **1/4 bit** | | **1/2 bit** | | **1 bit** | | **2 bits** |
| Generative code/math | | | | | | | | |
| human_eval_plu (pass@1) | 0.055 | 0.079 | +44.44% | 0.079 | +44.44% | 0.085 | +55.56% | 0.085 +55.56% |
| mbpp (pass@1) | 0.112 | 0.144 | +28.57% | 0.148 | +32.14% | 0.152 | +35.71% | 0.122 +8.93% |
| gsm8k (em) | 0.025 | 0.028 | +12.12% | 0.027 | +6.06% | 0.033 | +30.30% | 0.027 +6.06% |
| Multi-choice general knowledge / common sense | | | | | | | | |
| mmlu (macro_avg/acc_char) | 0.252 | 0.265 | +5.31% | 0.261 | +3.76% | 0.254 | +1.07% | 0.257 +2.19% |
| csqa (acc_char) | 0.199 | 0.175 | -11.93% | 0.199 | +0.00% | 0.187 | -6.17% | 0.197 -0.82% |
| hellaswag (acc_char) | 0.593 | 0.591 | -0.40% | 0.594 | +0.15% | 0.592 | -0.27% | 0.595 +0.32% |
| winogrande (acc_char) | 0.603 | 0.604 | +0.13% | 0.598 | -0.79% | 0.600 | -0.52% | 0.597 -1.05% |
| obqa (acc_completion) | 0.446 | 0.450 | +0.90% | 0.468 | +4.93% | 0.460 | +3.14% | 0.490 +9.87% |
| arc_challenge (acc_completion) | 0.400 | 0.392 | -1.93% | 0.386 | -3.43% | 0.405 | +1.29% | 0.385 -3.65% |
| arc_easy (acc_completion) | 0.596 | 0.602 | +0.92% | 0.592 | -0.64% | 0.603 | +1.06% | 0.592 -0.71% |
| piqa (acc_char) | 0.734 | 0.736 | +0.22% | 0.738 | +0.52% | 0.734 | +0.07% | 0.733 -0.15% |
| Multi-choice text understanding | | | | | | | | |
| race.high (acc_char) | 0.390 | 0.382 | -2.20% | 0.390 | +0.00% | 0.387 | -0.81% | 0.386 -1.03% |
| race.middle (acc_char) | 0.532 | 0.511 | -3.93% | 0.519 | -2.49% | 0.522 | -1.83% | 0.514 -3.40% |
| boolq (acc_completion) | 0.583 | 0.632 | +8.39% | 0.614 | +5.35% | 0.648 | +11.12% | 0.620 +6.29% |
| Culture | | | | | | | | |
| nq (em) | 0.081 | 0.069 | -15.36% | 0.073 | -9.56% | 0.075 | -7.17% | 0.071 -11.95% |
| tqa (em) | 0.205 | 0.191 | -6.93% | 0.190 | -7.58% | 0.200 | -2.84% | 0.197 -4.13% |

Table 2: Performance of 1.5B models trained on 47B tokens. The training procedure was tuned for the baseline and kept unchanged, but the Free Transformers require 3.6% more compute and parameters for the encoder. See Figure 4 in Appendix F for the performance during training.

| 8B models (200B tokens) | | | | | | | | |
|---|---|---|---|---|---|---|---|---|
| | **Baseline** | **Free Transformer** | | | | | | |
| | | 1/4 bit | | 1/2 bit | | 1 bit | | 2 bits |
| Generative code/math | | | | | | | | |
| human_eval_plu (pass@1) | 0.159 | 0.171 | +7.69% | 0.189 | +19.23% | 0.165 | +3.85% | 0.177 +11.54% |
| mbpp (pass@1) | 0.278 | 0.330 | +18.71% | 0.306 | +10.07% | 0.298 | +7.19% | 0.318 +14.39% |
| gsm8k (em) | 0.086 | 0.079 | -8.77% | 0.095 | +9.65% | 0.104 | +20.18% | 0.096 +10.53% |
| Multi-choice general knowledge / common sense | | | | | | | | |
| mmlu (macro_avg/acc_char) | 0.359 | 0.337 | -6.13% | 0.398 | +10.97% | 0.365 | +1.81% | 0.345 -4.00% |
| csqa (acc_char) | 0.356 | 0.292 | -17.93% | 0.450 | +26.21% | 0.346 | -2.99% | 0.324 -8.97% |
| hellaswag (acc_char) | 0.735 | 0.737 | +0.26% | 0.737 | +0.26% | 0.732 | -0.45% | 0.738 +0.39% |
| winogrande (acc_char) | 0.680 | 0.667 | -1.86% | 0.664 | -2.32% | 0.664 | -2.32% | 0.667 -1.86% |
| obqa (acc_completion) | 0.522 | 0.508 | -2.68% | 0.484 | -7.28% | 0.530 | +1.53% | 0.554 +6.13% |
| arc_challenge (acc_completion) | 0.465 | 0.483 | +3.87% | 0.468 | +0.55% | 0.452 | -2.95% | 0.485 +4.24% |
| arc_easy (acc_completion) | 0.677 | 0.676 | -0.25% | 0.665 | -1.81% | 0.668 | -1.44% | 0.679 +0.31% |
| piqa (acc_char) | 0.774 | 0.780 | +0.77% | 0.782 | +1.05% | 0.785 | +1.41% | 0.793 +2.46% |
| Multi-choice text understanding | | | | | | | | |
| race.high (acc_char) | 0.433 | 0.447 | +3.30% | 0.443 | +2.25% | 0.444 | +2.58% | 0.435 +0.53% |
| race.middle (acc_char) | 0.594 | 0.592 | -0.35% | 0.591 | -0.47% | 0.587 | -1.17% | 0.584 -1.64% |
| boolq (acc_completion) | 0.705 | 0.632 | -10.37% | 0.632 | -10.33% | 0.687 | -2.47% | 0.671 -4.82% |
| Culture | | | | | | | | |
| nq (em) | 0.181 | 0.183 | +1.38% | 0.167 | -7.67% | 0.173 | -4.14% | 0.168 -6.90% |
| tqa (em) | 0.440 | 0.438 | -0.28% | 0.443 | +0.80% | 0.434 | -1.19% | 0.446 +1.45% |

Table 3: Performance of 8B models trained on 200B tokens. The training procedure was tuned for the baseline and kept unchanged, but the Free Transformers require 3.1% more compute and parameters for the encoder. See Figure 5 in Appendix F for the performance during training.

## F    PERFORMANCE DURING TRAINING

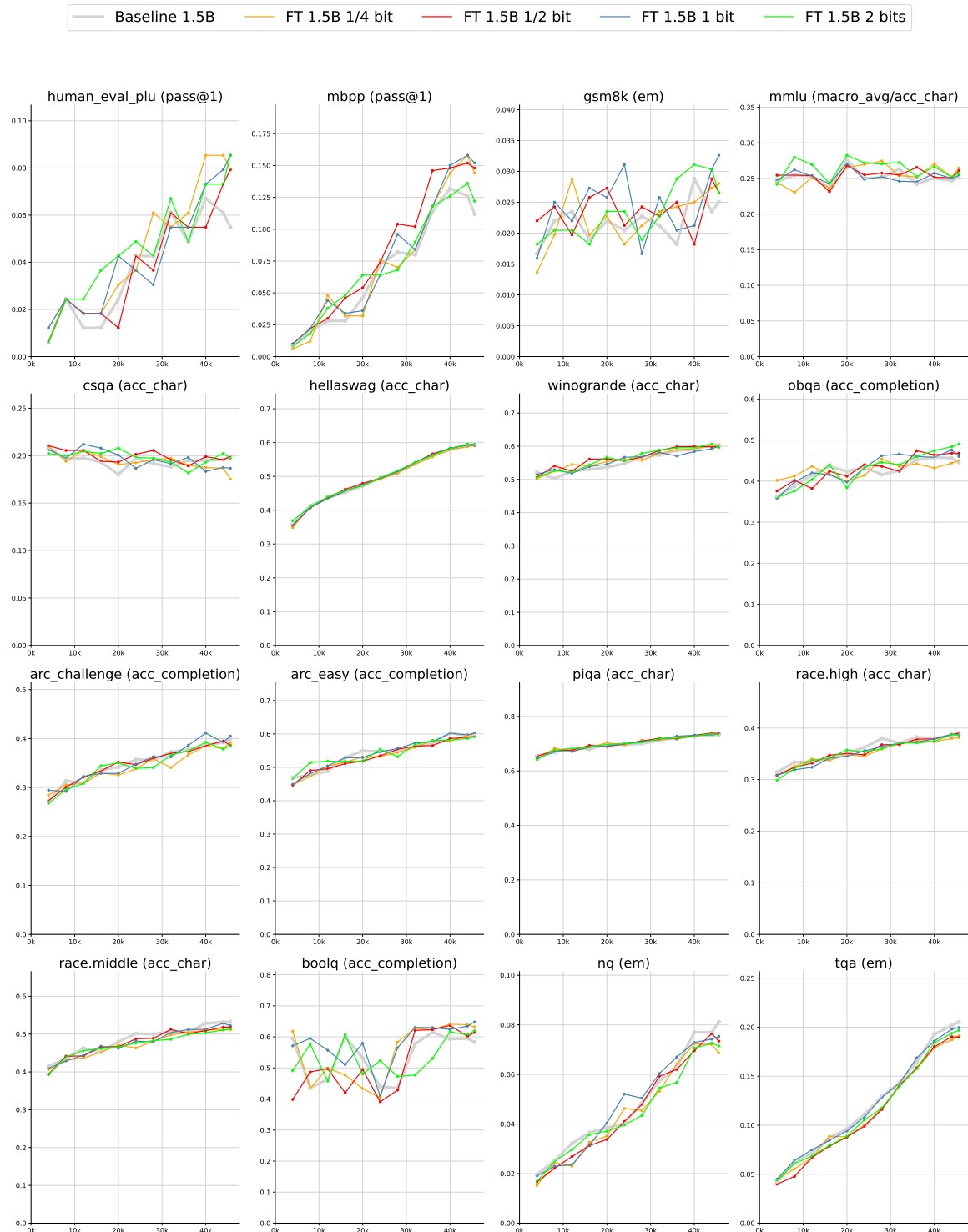

Figure 4: Experiments with 1.5B models trained on 47B tokens. Comparison on standard benchmarks of the baseline and our models. The optimization hyperparameters were tuned for the baseline and kept unchanged, but the Free Transformers require $3.6\%$ more compute and parameters for the encoder.

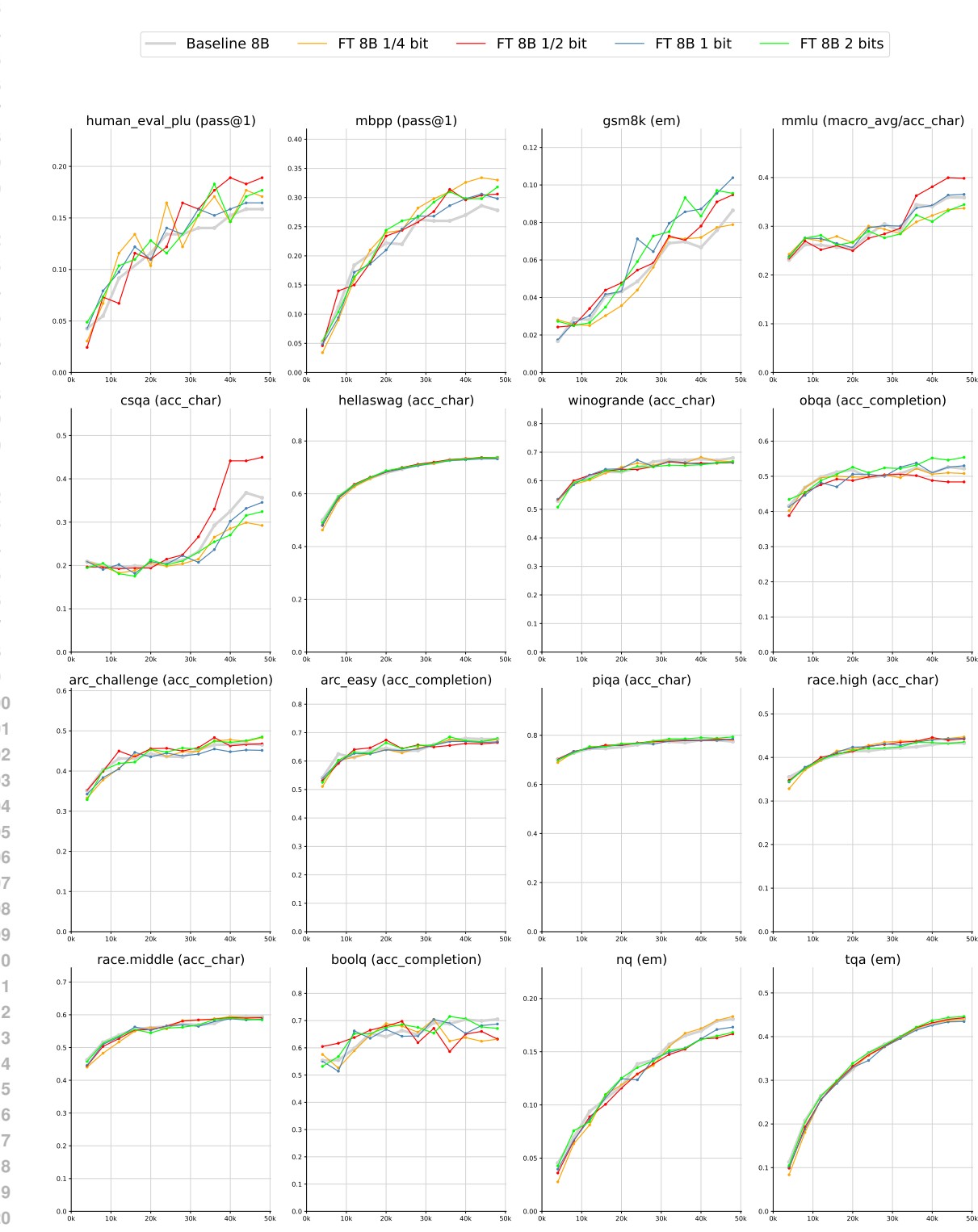

Figure 5: Experiments with 8B models trained on 200B tokens. Comparison on standard benchmarks of the baseline and our models. The training procedure was tuned for the baseline and kept unchanged, but the Free Transformers require 3.1% more compute and parameters for the encoder.

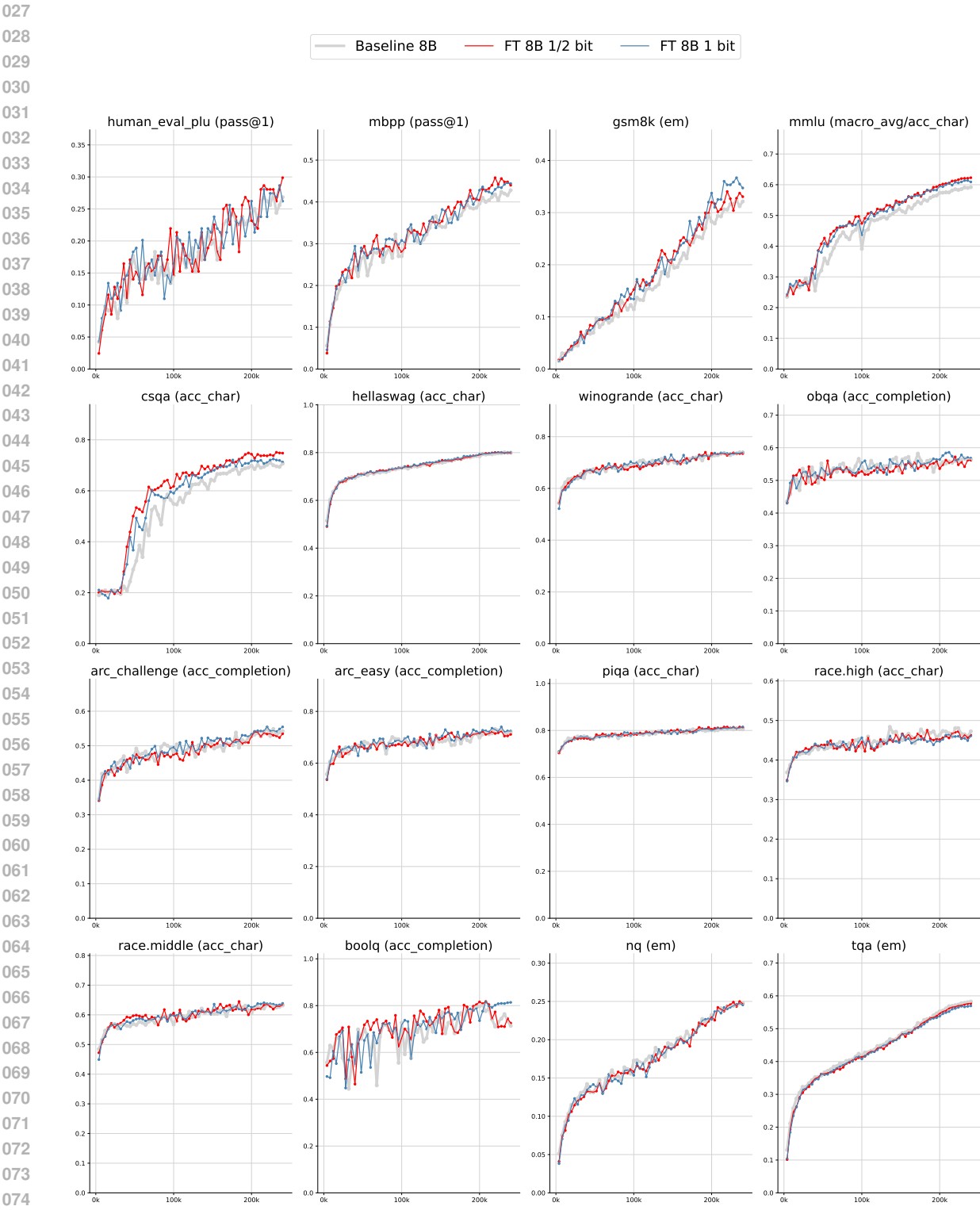

Figure 6: Experiments with 8B models trained on 1T tokens. Comparison on standard benchmarks of the baseline and our models. The training procedure was tuned for the baseline and kept unchanged, but the Free Transformers require $3.1\%$ more compute and parameters for the encoder.

