# OpenReview forum: "The Free Transformer"
_ICLR.cc/2026/Conference — Submitted to ICLR 2026_

### Official Review · Reviewer_cxJS · 2025-11-02

**Soundness:** 2
**Presentation:** 2
**Contribution:** 3
**Rating:** 2
**Confidence:** 4

**Summary:**

This paper proposes to do variational training for decoder-only Transformers. Specifically, the authors introduce a simple way to add some encoder layers (Fig 2) such that the compute from the decoder can be largely reused. The authors also proposed to replace the KL term in the ELBO with a "KL governor", which assigns smaller variance to the posterior if the posterior mean is close to 0, maintaining the KL term at a constant. The authors trained a 1.5B and a 8B model and compared with decoder-only Transformers in a set of reasoning benchmarks.

**Strengths:**

1. As argued by the authors, variational training of latent-variable LLMs is an interesting and largely under-explored topic.
2. The authors' proposal of latent inclusion and using KL governor to replace free bits in traditional VAEs are original.
3. The improvement over baselines "on the generative tasks that put less emphasis on syntax, and measure performance either through the validity of generated code when it is executed (HumanEval+ and MBPP) or through exact matches of the expected response, which requires a complex reasoning (GSM8K)" is inspiring.

**Weaknesses:**

1. This paper seems to be written in a rush, with critical technical details missing. For example, in Fig. 2, the input to the encoder seems to be the KV from the middle layer of decoder, and the Q from "a trained token embedding $\zeta$ replicated to match the sequence length" (L211). However, how would these embeddings constitute "Full Self-Att" with those KVs from input tokens given that they are not even the same set of tokens? Do these $\zeta$ have positional embeddings? Another example is that when introducing the details of the "KL governor", which is the key contribution of this paper, the authors only vaguely mention "There is no analytical form for this σ(∥µ∥2), and we need to propagate the gradient through it. So, given the value of the hyperparameter κ we solve it numerically for values of ∥µ∥2 and fit a degree-eight polynomial to it." (L261-263)

2. The empirical result is mixed, with minimum ablation missing. In the experiment section, the authors only reported the numbers on some reasoning benchmarks. Across all the free-bits value that the author swept over, none of the them outperforms baseline consistently. In fact, >5% degradation is very common in Table 1. IIUC, the baseline is just the autoregressive Transformer, without any RL-based thinking. In terms of ablation, the authors only briefly mentioned the proposed KL governor is more stable than directly optimizing KL (or its variant with free bits), without showing actual results. What if the whole training is deterministic and you still keep the clamp on $\mu$?

3. Critical related work missing: Phan et al. are probably the first to discuss the relation between latent reasoning and variational inference, who definitely deserve their credits. Kong et al. also introduced variational training to LLMs, which seems to be the most relevant existing work.

Phan et al., Training chain-of-thought via latent-variable inference, NeurIPS'23

Kong et al., Latent Thought Models with Variational Bayes Inference-Time Computation, ICML'25

**Questions:**

The authors pretrained the proposed model, what is the perplexity?

The reported mixed results seems to imply a zero-sum game (at a cost of >5% extra training compute), how would the authors postulate the generic usage of the proposed method?

---

> ### Author Response · Authors · 2025-11-18
> **Response to Reviewer cxJS**
>
> Indeed a positional embedding (RoPE, section 4.1) is added in every block to zeta. We will state it explicitly in 3.2
>
> We do not think that the "KL governor" is the key contribution of the paper, and regarding the polynomial approximation, the problem is that if we fix kappa and mu, there is a resulting sigma, but it does not have an analytical expression, which is problematic compute-wise if we do it numerically every time we compute sigma during training, and also to use it with autograd. So we compute sigma numerically for many mus when we create the model, and fit a polynomial. We then use this polynomial during training, which is fast to compute and autograd-friendly. We will clarify the paper.
>
> We hope that the experiments with 1T tokens described in the common response address the empirical result remarks.
>
> Thanks for the references about latent reasoning and variational inference.
>
> We unfortunately did not compute validation perplexity for the models in the paper.
>
> The zero sum game remark is unclear, what do you mean?

---

### Official Review · Reviewer_DgxJ · 2025-11-02

**Soundness:** 3
**Presentation:** 2
**Contribution:** 2
**Rating:** 2
**Confidence:** 4

**Summary:**

The paper proposes a “Free Transformer,” a decoder-only transformer that is trained as a conditional VAE but keeps inference almost identical to a normal decoder. The author used two non-causal encoder layers to produce a tokenwise Gaussian posterior, and forces the KL between this posterior and a fixed standard normal prior to be a pre-chosen value. The experiment part provides 2 size of the models performance on several benchmarks.

**Strengths:**

I really appreciate the author for the simple and smart design, which includes minimal changes to the architecture. And according to the paper, the training and inference time doesn't affect a lot by the latent design.

The model has a clear and explicit latent variable to control the generation, or probably perform some kind of reasoning in latent space. I think this is a great approach and "reasoning in the latent space" is a promising direction.

**Weaknesses:**

1. Model-related issue
* A big autoregressive decoder induces a complex, non-Gaussian posterior, but the paper uses only 2 non-causal layers to output a Gaussian posterior. This is a very narrow variational family and likely leaves an amortization gap.
* I understand the author want to solve this posterior collapse issue using fixed KL. However, this approach must spend the same KL budget for all samples, so easy examples get unnecessary noise and hard examples cannot request more bits. This directly limits test-time reasoning on harder instances and is tightly coupled to the train–test latent mismatch, which contradict to the idea of this paper of adapting reasoning in different cases via latent variables.
* Training always feeds structured, sequence-conditioned z at a fixed KL while the inference always feeds unconditioned N(0,I). In my understanding, in conditional generation, posterior should be used instead of prior. The paper does not show how much performance or diversity lost because of this. I hope the author can explain more on this part.

2. Experimental concerns
* This part is my another major concern due to limited results and details/design included in the paper.
* Since the author is studying a generative model, then density estimation and perplexity comparison should be included in the experiments. A basic validation perplexity of baseline GPT and other variational/diffusion based methods should be included. Because the method is based on ELBO, it is easy to see the comparison between different baselines in conditional and unconditional generation tasks. Without this, we cannot tell whether the method actually improves language modeling.
* For the training data, the experiments specify only the token counts (47B, 200B) and the hardware, but not the underlying data distribution (sources, domain mix, code ratio, math ratio, filtering). Since the reported gains are largest on code, math, and CSQA, this is a confound. The authors should either (i) describe the training corpus in detail, or (ii) run ablations where the baseline and the Free Transformer are both trained on an explicitly defined subset (e.g. pure text, pure code) to show that the latent is the cause of the improvement, not the data mixture.
* There is no concrete numbers to measure the training/inference efficiency. The paper claims “about 6–7% extra cost,” but it does not show per-token TFLOPs, tokens/sec, or memory usage compared to the baseline. Because the main selling point is “latent for free,” this should be quantified.
* Also, more experiments like prior sampling and diversity results should also be added. The paper does not show this, so we cannot judge whether the learned latent space produces diverse, coherent generations.
* More analysis on the behavior of this model should be studied. For example, I'm curious about which layers for adding z plays the most important change to the generation. Also, the author didn't show any results of changing z value could lead to different generation, especially the paper has a discussion of examples in the "motivation" section. These should be easily implemented in the ablation study.

**Questions:**

1. The encoder design is very simple, so I'm curious if the adding more layers to the encoder can bring better performance. From test-time scaling perspective, I'm expecting to see which factor of the design is important for reasoning capability. Is it still the model size?  If so, then the scaling behavior is the same as GPT based method. Are there any scaling in the latent space?
2. I'm really appreciate if the author can provides more details and experiments based on what I mentioned in the Weakness part.
3. In real practice, how does the KL setting determined. I'm trying to understand if this is the bottleneck of this method.
4. Is line 179 a typo?

---

> ### Author Response · Authors · 2025-11-18
> **Response to Reviewer DgxJ**
>
> You point out issues with the models with which we agree and we are currently investigating (e.g. per-sample kl). However since science is a never ending sequence of improvements, "it could be better" would virtually reject any publication in a technical field.
>
> Regarding the experimental concerns, we hope that the additional experiments described in the common response address the issue.
>
> Regarding the overhead, we are adding two transformer blocks for the encoder, and the baselines have originally 28 for the 1.5B model and 32 for the 8B. The overhead comes from the proportionality between flops / wall clock and number of blocks.
>
> Regarding the number of blocks in the encoder, our initial designs had a full encoder (hence x2 compute) and were performing very well, but we scrapped those designs as we wanted to avoid a tricky discussion with reviewers about the overhead.
>
> The KL is fixed as a hyper parameter.
>
> Yes line 179 is a typo, it should be Q(Z | S = s), thanks.

---

### Official Review · Reviewer_yz81 · 2025-11-03

**Soundness:** 3
**Presentation:** 2
**Contribution:** 2
**Rating:** 4
**Confidence:** 5

**Summary:**

The paper proposes the "Free Transformer," a VAE-based decoder that conditions its generation on a latent variable $Z$. Its core novelty is the "KL Governor," a training mechanism that forces the $D_{KL}(Q(Z|S) || P(Z))$ to a fixed target value $\kappa$ rather than adding it as a loss penalty. This is intended to prevent posterior collapse and, by discarding the encoder at inference, adds no computational overhead. The authors report strong gains on generative tasks like HumanEval+ and GSM8K.

**Strengths:**

1. Motivation: The core idea is compelling: enabling models to use explicit latent "plans" rather than relying on purely "post-hoc" autoregressive token-level decisions.

2. Efficiency: The design is highly practical, incurring a small training overhead and zero inference-time cost.

3. Novelty: The "KL Governor" is a creative and new approach to managing the notoriously unstable VAE training objective.

4. Empirical Signals: The strong performance gains on complex coding and math reasoning tasks are significant and warrant attention.

**Weaknesses:**

The paper's central claim, while promising, rests on assumptions that would be significantly strengthened by further validation.

1. The Decoder is Not Constrained, and Its Usage of $Z$ is Unproven.

The paper's core thesis rests on the decoder using the latent $Z$. However, the proposed "KL Governor" only constrains the encoder to produce an informative $Z$; it does not place any direct constraint on the decoder. This leaves a critical question unanswered: it is unclear if the powerful autoregressive decoder truly learns to depend on $Z$ or if it simply relies on its own token-level context, effectively ignoring $Z$. This is a key aspect of the posterior collapse problem this paper aims to solve.

To validate this core claim, a critical ablation is necessary: an inference-time test with the latent variable disconnected (e.g., $Z=0$). If the decoder is dependent on $Z$, performance should collapse. Without this experiment, it is difficult to be certain that the reported gains are not just a side-effect of the VAE structure acting as a complex training-time regularizer, rather than the decoder actively using the latent "plan" at inference.

2. Inconsistent Performance and Unaddressed Regressions.

The performance gains appear to be highly task-specific. The paper highlights wins on generative tasks but does not address the notable performance regressions on several standard benchmarks (e.g., -8.46% on boolq and -5.21% on nq for the 8B model). A discussion of this trade-off—why the model might improve on reasoning while regressing on other tasks—is needed to fully interpret the results.

3. Unexplored Design Choices.

The paper introduces several specific design choices without justification or comparison.

(1) KL Governor vs. Alternatives: The "KL Governor" is novel, but the paper would be stronger if it were benchmarked against standard KL regularization methods like $\beta$-VAE or KL annealing, which also aim to prevent posterior collapse.

(2) Architectural Choice: The decision to inject $Z$ only at the middle layer is presented without an ablation or justification. It is unclear why this specific location was chosen over, for example, the input layer or all layers.

**Questions:**

See above.

---

> ### Author Response · Authors · 2025-11-18
> **Response to Reviewer yz81**
>
> We hope that the additional experiments described in the common response address the--very legitimate--questions about the usage of Z and provide solid evidence of the performance.
>
> Regarding the "Unexplored design choices", we can indeed probably do more than training four 8B parameters models from scratch on 200B tokens, but that sets the bar pretty high in terms of compute budget.

---

### Official Review · Reviewer_Sddi · 2025-11-04

**Soundness:** 2
**Presentation:** 3
**Contribution:** 3
**Rating:** 4
**Confidence:** 5

**Summary:**

This paper proposes an LLM with one latent vector per token as sources of variation added to a middle layer of an otherwise-standard decoder-only transformer.  As in a VAE, at training time an encoder finds high probability samples of these variables conditioned on the observed text.  Several interesting architectural proposals improve the training efficiency and effectiveness of this model, including sharing the initial layers of the encoder with the initial layers of the decoder, and fixing the KL divergence rather than learning to minimise it.  Empirical evaluations on a large number of LLM benchmarks show promising results, suggesting that adding variance to internal representations of the transformer has added value over just adding variance to the token prediction step.

**Strengths:**

This paper contributes interesting insights into latent variable models for LLMs and a collection of effective methods for making the latent variables work as random choices about future outputs.  Simply adding randomness to hidden representations would not have the same effect, since during LM training these random choices would be independent of future outputs.

A central feature of this latent variable model is that there is one latent vector per token, which is novel with respect to the previous LLM models reviewed in this paper, but related to the work on nonparametric VAEs (Henderson and Fehr, ICLR 2023).  Combined with their method for fixing the KL divergence per token, this means that the number of bits conveyed by the latent variables grows linearly with the length of the generated text, which is essential for modelling the variability in language.

A large number of evaluations are run on reasonably-sized models trained from scratch.  These results succeed in showing the potential of this novel new direction for LLM architectures.

**Weaknesses:**

The paper seems to be work in progress.  There is very little discussion of the empirical results, and no ablation studies other than a standard LLM baseline.  The description of the model misses some key points (see below).

The experiments don't seem to be testing any hypothesis.  The paper reads like they have a cool idea, so lets see what happens. The conclusion is that something interesting happens, but it is not clear what.  This impression is reinforced by the lack of any ablation studies.

As made clear in Figure 1, there is one latent vector per token.  But Figure 1 is never referenced in the text, and nothing in the specification of the model mentions this fact.  They even say that the prior is "standard Gaussian noise", but Gaussians are parametric distributions, while here the number of sampled values grows with the number of tokens in the text.  This is at best a misleading specification, and I would say incorrect.  In particular, the variable names violate standard naming conventions for vectors versus matrices and tensors.

**Questions:**

Please explain how Figure 1 is related to the specification in Section 3.  What are your variable naming conventions?

Please add the "batch size" to Figure 2.  The answer cannot be "we omit batch size", since this is an essential novelty of your approach over previous latent-variable LLMs.

---

> ### Author Response · Authors · 2025-11-18
> **Response to Reviewer Sddi**
>
> The hypothesis we test is if adding a variational component to allow random latent conditioning improves performance, and it does.
>
> Regarding the one latent vector per token, it is indeed true that the text is really not clear, apologies. We will fix this in the camera ready.
>
> With z_t from Figure 1 and Z from section 3, we have Z=(z_1, ..., z_T). And x^l_t in Figure 1 is the activation computed after layer l. But while correct it is true that the arrow between z_t and x^3_t in Figure 1, which means "z_t modulates x^3_t by being added to the KV input of the transformer block computing x^3_t" should be explained more clearly. We will fix it.
>
> Regarding the batch size in the caption of Figure 2, we realized that the submitted pdf lacks the tensor sizes, which makes the caption very confusing. In any case, "batch size" here is not the number of tokens, but the number of sequences processed in parallel, which is irrelevant to the understanding of the processing. Technically, our implementation uses a dynamic masking to deal with batches of sequences of variable lengths without padding, but this goes beyond the scope of the paper. We will fix Figure 2 by adding back the tensor sizes, sorry for that.

---

### Author Response · Authors · 2025-11-18
**Common response to the reviewers**

We thank the reviewers for the time spent on our paper and their comments.

Some of the negative remarks come from a concern that this paper does not provide tangible scientific results. We strongly disagree with this perspective. The community suffers from contributions that are ad hoc and complex recipes, with hyper-parameter overtuning, tested on exotic data sets and performance measures.

Having this in mind, we were extremely careful to (1) simplify our method as much as possible and formulate it as a classic VAE (2) keep absolutely unchanged the optimization procedure and training data used for the baselines (3) use standard benchmarks and (4) compare to very strong baselines.

This setup assures that this method improves the performance, and its simplicity makes it a valuable candidate for future extensions and theoretical analysis.

If we had a "cool idea" (as phrased by reviewer Sddi) to improve a very strong llama-3 with a sound VAE extension, it is unclear why that would be grounds for rejection. And the reality is that this model is the result of arduous research work. "cool ideas" for architecture improvement do not get +5% on a llama-3 8B.

At the time of writing this rebuttal, we have run an additional set of experiments (with the model now using a single non-causal block for the encoder, reducing the overhead from ~6% to ~3% and a binary discrete Z with STE instead of the Gaussian Z). The numbers presented in Table 1 overall improve slightly.

We also trained a 8B with 1T tokens and get as relative improvement (averaged on the last third of the training steps to filter out the noise) +4.22% on human eval, +6.08% on mbpp, +5.84% on gsm8k, +5.16% on mmlu, +6.28% on csqa, the only degradation below -1% is race.high at -1.55% all other scores are between -1% and +1%.

Finally, we investigated the latent representation with a small free transformer and synthetic sequences composed of a character repeated 64 times and a "target" made of a random letter repeated 8 times at a random location. Experiments show that if train models with a KL gradually increasing from zero, and generate several sequences with the same Z, the latent representation encodes initially nothing, then the position of the target, then that position and the noise, and finally too much information leading to a collapse of the generative process.

All these experiments will be added to the final version.

---

> ### Author Response · Authors · 2025-12-02
>
> There has been unfortunately no reaction to our rebuttal, and our understanding is that further comments--like this one--are for the newly assigned AC only.
>
> TL;DR: We present a clear and principled variational extension of a decoder transformer and get 5%+ on humaneval+, mbpp, gsm8k, mmlu, and csqa, without degradation on others compared to a sotal 8B llama-3 with *exactly the same parameter count, training setup, and data mix*.
>
> This paper presents a straightforward variational extension of the standard decoder transformer, and validates its performance with 1.5B models and 8B model.
>
> Beside heavy rewriting and an updated method that reduces the training overhead by half (~6% -> ~3%), the updated version addresses the reviewers' concerns with (1) an experiment with synthetic data demonstrating that the latent state indeed encodes a meaningful representation (2) experiments with a 8B model trained on 1T tokens that confirm the performance at that scale.
>
> We again insist on the fact that (1) our method is sound and conceptually clear (2) we used an unmodified setup for optimization and datamix used for the sota llama3 8B baseline, and (3) we gets 5%+ on important benchmarks.
>
> If that is not enough, what is needed to get into ICLR is confusing at this point.

---

### Meta-Review · Area_Chair_y4yN · 2026-01-05

**Summary:**

I agree with the reviewers that the idea of the paper is simple, elegant, and appealing. Adding a latent variable to the decoder with very small overhead is attractive, and the paper shows improvements on several benchmarks. In this sense, the work is a tangible contribution and shows promising results.

However, the main issue is not the idea itself, but how the reviewers’ concerns are handled. Many reviewers raised legitimate and important questions, but the rebuttal often responds with very limited detail. Instead of clearly explaining where and how each concern is addressed, the authors frequently refer back to the paper or to added experiments without enough explanation. There is no in-depth discussion of the many experimental results, including those in the appendix. It is often unclear which results are meant to address which reviewer comments. For this reason, I do not think any reviewer would significantly change their assessment of the paper based on the current response.

There are also many open questions about the method, and many of them do not require training larger LLMs to be answered. Key design choices are not analyzed, such as the effect of the layer where the latent variable is injected, or whether the method could be used to reduce the number of layers or attention heads without degrading performance. Important evaluations are missing, including perplexity results, ELBO analysis for different variational families, and diversity analysis. Without these experiments, it is hard to judge whether the learned latent space produces diverse and coherent generations.

In addition, reviewer cxJS pointed out two important related works by Phan et al. and Kong et al., which are not clearly discussed or connected to the proposed method in the revised paper.

Overall, while I greatly appreciate the idea and the potential of the approach, the paper is not ready for acceptance in its current form. I encourage the authors to carefully incorporate the reviewers’ feedback and submit a stronger and more complete version to a future ML venue.

**Reviewer Concerns:**

**Reviewer Concerns**

**Addressed by the rebuttal**
- Clarified some missing technical details in the model description and figures.
- Added additional experimental results on larger models and longer training.
- Provided partial clarification on the role of the latent variable and the KL governor.
- Fixed some presentation issues and typos noted by reviewers.

**Still outstanding**
- Lack of clear and structured responses explaining how each reviewer concern is addressed.
- Missing in-depth analysis of experimental results, including those in the appendix.
- No ablation on where the latent variable is injected in the transformer.
- No study on reducing layers or attention heads using the proposed method.
- Missing perplexity results.
- Nodiversity analysis of the learned latent space.
- Mixed results across tasks are not properly discussed.
- Missing or insufficient discussion of key related work (e.g. Phan et al., Kong et al.).

**Reviewer Scores:**

I do not think any of the reviewers would have substantially modified their scores. Given the low level of detail in the rebuttal and the lack of clear responses to the main concerns, reviewers would likely maintain their original assessments.

---

### Decision · Program_Chairs · 2026-01-26

Reject